# The Diversity and Functional Capacity of Microbes Associated with Coastal Macrophytes

Khashiff Miranda,[a] Brooke L. Weigel,[b,c] Emily C. Fogarty,[d,e] Iva A. Veseli,[f] Anne E. Giblin,[g] A. Murat Eren,[d,h] Catherine A. Pfister[b,i]

[a]The College, The University of Chicago, Chicago, Illinois, USA
[b]Committee on Evolutionary Biology, The University of Chicago, Chicago, Illinois, USA
[c]Friday Harbor Laboratories, University of Washington, Friday Harbor, Washington, USA
[d]Department of Medicine, The University of Chicago, Chicago, Illinois, USA
[e]Committee on Microbiology, The University of Chicago, Chicago, Illinois, USA
[f]Biophysical Sciences Program, The University of Chicago, Chicago, Illinois, USA
[g]The Ecosystems Center, The Marine Biological Laboratory, Woods Hole, Massachusetts, USA
[h]Josephine Bay Paul Center, Marine Biological Laboratory, Woods Hole, Massachusetts, USA
[i]Department of Ecology & Evolution, The University of Chicago, Chicago, Illinois, USA

**ABSTRACT** Coastal marine macrophytes exhibit some of the highest rates of primary productivity in the world. They have been found to host a diverse set of microbes, many of which may impact the biology of their hosts through metabolisms that are unique to microbial taxa. Here, we characterized the metabolic functions of macrophyte-associated microbial communities using metagenomes collected from 2 species of kelp (*Laminaria setchellii* and *Nereocystis luetkeana*) and 3 marine angiosperms (*Phyllospadix scouleri*, *P. serrulatus*, and *Zostera marina*), including the rhizomes of two surfgrass species (*Phyllospadix* spp.), the seagrass *Zostera marina*, and the sediments surrounding *P. scouleri* and *Z. marina*. Using metagenomic sequencing, we describe 63 metagenome-assembled genomes (MAGs) that potentially benefit from being associated with macrophytes and may contribute to macrophyte fitness through their metabolic activity. Host-associated metagenomes contained genes for the use of dissolved organic matter from hosts and vitamin ($B_1$, $B_2$, $B_7$, $B_{12}$) biosynthesis in addition to a range of nitrogen and sulfur metabolisms that recycle dissolved inorganic nutrients into forms more available to the host. The rhizosphere of surfgrass and seagrass contained genes for anaerobic microbial metabolisms, including *nifH* genes associated with nitrogen fixation, despite residing in a well-mixed and oxygenated environment. The range of oxygen environments engineered by macrophytes likely explains the diversity of both oxidizing and reducing microbial metabolisms and contributes to the functional capabilities of microbes and their influences on carbon and nitrogen cycling in nearshore ecosystems.

**IMPORTANCE** Kelps, seagrasses, and surfgrasses are ecosystem engineers on rocky shorelines, where they show remarkably high levels of primary production. Through analysis of their associated microbial communities, we found a variety of microbial metabolisms that may benefit the host, including nitrogen metabolisms, sulfur oxidation, and the production of B vitamins. In turn, these microbes have the genetic capabilities to assimilate the dissolved organic compounds released by their macrophyte hosts. We describe a range of oxygen environments associated with surfgrass, including low-oxygen microhabitats in their rhizomes that host genes for nitrogen fixation. The tremendous productivity of coastal seaweeds and seagrasses is likely due in part to the activities of associated microbes, and an increased understanding of these associations is needed.

Address correspondence to Khashiff Miranda, khashiff.miranda.1@ulaval.ca.

The authors declare no conflict of interest.

10.1128/msystems.00592-22 **1**

**KEYWORDS** host-microbiome relationships, kelp, macrophytes, marine microbiology, oxygen, seagrass, surfgrass

We are experiencing a paradigm shift in biology with the recognition that many species exist as a consortium with microbes (1). These microbial associations are nearly ubiquitous, spanning a diversity of hosts across ecosystems. In coastal marine environments, marine angiosperms and macroalgae show high productivity (2), are a critical component of the carbon (3) and nitrogen cycles (4, 5), and host a wide range of microbial organisms. Different macroalgal species (6, 7) and their different tissues (8, 9) host distinct microbial communities which number in the millions per $cm^2$ of host tissue (10). Yet, we still know little about the functional role the microbiome plays in host fitness and how the host influences the microbiome. Indeed, the contributions that marine macrophytes make to global carbon and nitrogen cycling have largely ignored the roles that microbes play. Yet, bacteria can supply B vitamins (11) and affect the development of their seaweed hosts (12). Seagrasses have been shown to have microbial associates that help the host access inorganic nitrogen, including bacteria that can ammonify (13) and bacteria that can fix atmospheric nitrogen (5). Though many seaweeds and seagrasses are foundational species in the coastal ocean, our understanding of the diversity and roles of their associated microbes is nascent, even as we discover that environmental change affects these communities (14).

Marine foundational species influence their surrounding physical environment. Macrophytes alter oxygen gradients and potentially influence the associated microbial communities across a range of spatiotemporal scales. The photosynthetic and respiratory activities of the host can generate a "phycosphere" (15) in which the host influences the physical environment experienced by the microbes, sometimes over micron or mm scales. For example, the basal leaf meristem of the seagrass *Zostera* ranges from well-oxygenated to anoxic over a scale of 300 $\mu$m (16). At larger scales, canopy forming kelps elevate daytime dissolved oxygen (DO) levels in seawater within kelp forest stands compared to the surrounding water column (17). At night, however, the DO levels in these kelp forests may drop to between 2 to 10 mg $L^{-1}$ (18). Similarly, diel fluctuations that range an order of magnitude have been measured in rocky shore tidepools that include the surfgrass *Phyllospadix scouleri* (19, 20). This range of oxygen concentrations likely generates temporal partitioning of oxic and anoxic metabolisms and enhances the diversity of the microbial metabolisms that may be associated with macrophytes.

In low-oxygen environments, microbial respiration with alternative terminal electron acceptors, such as nitrate or sulfate, is common. Kelp (21, 22) and surfgrasses (23, 24) release >10% of their fixed carbon as dissolved organic carbon (DOC), generating a reservoir of labile carbon in the nearshore water column. Sulfur and nitrogen reducing microbes may respire these complex carbon substrates anaerobically while in anoxic environments, altering nutrient availability in the process. The subsequent replenishment of these consumed terminal electron ($SO_4^{2-}$ and $NO_3^-$) acceptors through sulfur and nitrogen oxidizing metabolic pathways, such as *sox*-sulfur oxidation (25) and nitrification, (26) allows these microbial metabolisms to persist. While these biogeochemical processes are well-established, little is known about microbial nutrient cycling associated with marine macrophytes, such as kelp or seagrasses.

In coastal systems, nitrogen can limit primary production, and microbial associates that aid in accessing nitrogen might be beneficial. Oxygen fluctuations can result in nitrogen oxidation or reduction (19). In proximity to the host, this retention of nitrogen may improve host fitness. Additional microbial metabolisms that can increase the available dissolved inorganic nitrogen (DIN) for the host (13) include pathways that cleave carbon-nitrogen bonds to generate ammonium. This ammonification in biological systems can result from a diversity of hydrolases, including ureases and other enzymes that cleave C-N bonds (27). Further, microbes that fix atmospheric nitrogen have been discovered in an increasing number of taxa (28, 29), now recognized to include both

heterotrophic and phototrophic taxa (30–32). Nitrogen fixation was previously assumed to be restricted to nitrogen-poor environments, but it has been quantified recently in systems thought to be nitrogen-rich (28, 33). This is an enigmatic finding, given that nitrogen fixation is a costly metabolic process that consumes 16 ATPs per $N_2$ fixed (34). Sediments where oxygen and nutrients are low, such as the rhizosphere of seagrasses, have provided evidence of nitrogen fixation (35–39). The recent discovery that nitrogen fixation takes place on particles in the coastal ocean where nitrate is relatively abundant (28, 33) suggests that *nifH* genes could be abundant in other nearshore systems.

Microbial metabolisms that synthesize compounds and vitamins needed by seaweeds and seagrasses may also underlie host-microbe exchanges. The active form of Vitamin $B_1$ (thiamine) is essential for all organisms and is involved in carbohydrate and amino acid metabolism. Vitamin $B_2$ (riboflavin) is integrated with co-enzymes in various oxidases ais and are involved in photosynthesis and phototropism (40). Vitamin $B_7$ (biotin) is a cofactor for acetyl coenzyme A (coA) which is essential for fatty acid synthesis. Vitamin B12 (cobalamin) is required as a coenzyme in the mitochondria, yet many algae depend upon prokaryotes to produce it (11, 41). Thus, marine macrophytes may be auxotrophic for key vitamins, and their production by host-associated bacteria may be another basis for species interactions in nature.

Here, we analyzed microbial metagenomes collected from 5 different coastal macrophytes to determine if there is functional genomic evidence of microbial metabolisms that could reciprocally benefit hosts and microbes. We examined microbial taxa and metabolisms in multiple microhabitats in which macrophytes alter the oxygen environment, including the inner bulb of the canopy kelp *N. luetkeana*, hypothesizing that this relatively low-oxygen environment, which is also high in carbon monoxide and nitrogen gas (42), could have unique microbial metabolisms. We also hypothesized that the rhizomes of the seagrasses *P. scouleri and P. serrulatus*, as well as the associated sediment around *P. scouleri*, would be low oxygen environments based on their ability to trap sediment in this otherwise high-energy rocky shore environment. We analyzed the microbiome of the more oxygenated surface of the blades of *P. scouleri* and *Laminaria setchellii*, comparing them with metagenome-assembled genomes (MAGs) from blades of *N. luetkeana* that were sampled on the same day in the same area for a separate study (43). We sampled the microbial community from the *Zostera marina* rhizosphere from West Falmouth Harbor, MA, USA, an environment that is well-characterized (44, 45).

We analyzed the microbial taxa present and examined their gene content to estimate functional and metabolic capacities. We hypothesized that microbial partners: (i) enhance host access to dissolved inorganic nitrogen through nitrogen recycling, ammonification, and nitrogen fixation, (ii) provide vitamins $B_1$, $B_2$, $B_7$, $B_{12}$, and (iii) use a diversity of abundant dissolved organic carbon exudates from the host. We tested whether microbial taxonomy and function differed across hosts and host tissue types as well as whether anaerobic metabolisms were present in the low-oxygen environments found within rhizomes and the surrounding sediment. We found that the range of oxygen environments engineered by host macrophytes likely explains the diversity of oxidizing and reducing microbial metabolisms and contributes to the diverse functional capabilities of microbes and their influences on carbon and nitrogen cycling in nearshore ecosystems.

## RESULTS

**Surfgrass rhizomes have lower oxygen concentrations than surrounding seawater.** The oxygen environment in the rhizomes differed significantly from that of the surrounding seawater (Fig. 1). At near-peak photoperiod, the sediment in the rhizosphere maintained a lower dissolved oxygen (DO) concentration than did the surrounding seawater for both *P. scouleri* ($n = 18$, pairwise $t$-test: $P < 0.001$) and *P. serrulatus* ($n = 11$, pairwise $t$-test: $P < 0.001$). *P. serrulatus* maintained a slightly lower DO concentration in the rhizome at 2.11 mg $L^{-1}$, compared to 5.61 mg $L^{-1}$ for *P. scouleri*. However, sampling likely introduced oxygenated water from the surrounding water column to the rhizome-

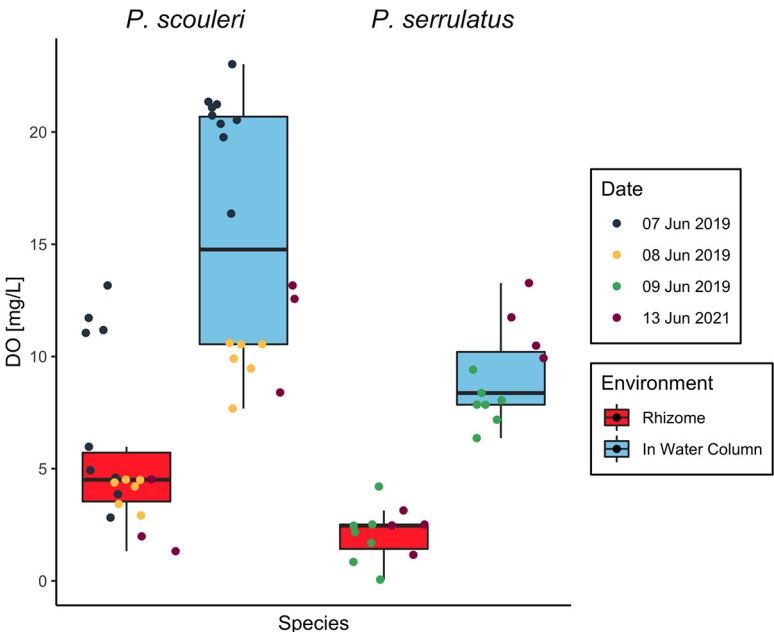

**FIG 1** Boxplot comparing the dissolved oxygen concentrations of macrophyte tissue in the water column (blue) and in the sediment-rhizome environment (red) of *P. scouleri* (pairwise *t*-test: $P < 0.001$) and *P. serrulatus* (pairwise *t*-test: $P < 0.001$). Sampling dates are represented by different colors.

sediment microenvironment, suggesting that the actual DO concentrations within the sediment were lower than the values reported.

**Diversity of MAGs assembled across hosts.** Following quality control and filtering, we obtained an average of 41 million sequence reads per sample (range 6.48 to 67.73 million) with 79.8% of raw reads retained on average. When these metagenomic short reads were assembled into contigs of at least 1,000 nucleotides, a mean of 42,026 contigs and a mean of 110,054 genes were present across samples (per sample summary provided in Table 1).

Across 8 metagenomes, we assembled 30 high quality MAGs, defined as having a completion score of >90% and a redundancy (or contamination) of < 10% (Table 2), using the criteria of Bowers et al. (46). We also identified 33 medium quality MAGs that had completion scores between 42% and 90% and redundancy scores between 0% and 11% (Data Set S1, Sheet 2). All MAGs were bacterial except for a single archaeon, *Crenarchaeota*, from the rhizome of *P. scouleri*. The bacterial MAGs spanned 8 phyla, including *Proteobacteria* ($n = 34$), *Bacteroidota* ($n = 17$), *Verrucomicrobiota* ($n = 2$), *Campylobacterota* ($n = 3$), *Desulfobacterota* ($n = 5$), and a single MAG in each of *Desulfuromonadota*, *Acidobacteriota*, and *Spirochaetota*. There were 43 MAGs resolved to the species level, with 14 to the genus level, 9 to the family level, 2 to the order

**TABLE 1** Summary of the features of 8 metagenomes. More information is in Data Set S1, Sheet 1, and the genome taxonomy is in Data Set S1, Sheet 2

| *Phyllospadix scouleri* | | | *Phyllospadix serrulatus* | *Laminaria setchellii* | *Nereocystis luetkeana* | *Zostera marina* | |
|---|---|---|---|---|---|---|---|
| Sediment | Rhizome | Blade | Rhizome | Blade | Inner bulb | Sediment | Rhizome |
| Number of quality reads (in millions) | | | | | | | |
| 43.68 | 67.73 | 38.41 | 37.99 | 48.58 | 6.48 | 19.37 | 65.76 |
| Bacteria | | | | | | | |
| 63.7% | 58.3% | 63.6% | 63.0% | 63.9% | 62.2% | 33.6% | 60.6% |
| Archaea | | | | | | | |
| 34.2% | 38.3% | 33.1% | 33.3% | 32.7% | 35.9% | 62.1% | 34.2% |

**TABLE 2** Metagenome-assembled genomes across all samples and their representation across phyla. More detailed information on the MAGs can be found in Data Set S1, Sheet 2

| *Phyllospadix scouleri* | | | *Phyllospadix serrulatus* | *Laminaria setchellii* | *Nereocystis luetkeana* | *Zostera marina* | |
|---|---|---|---|---|---|---|---|
| Sediment | Rhizome | Blade | Rhizome | Blade | Inner bulb | Sediment | Rhizome |
| High Quality MAGs | | | | | | | |
| 2 | 1 | 6 | 6 | 9 | 1 | 2 | 3 |
| Other MAGs | | | | | | | |
| 2 | 2 | 8 | 7 | 7 | 0 | 5 | 8 |
| Proteobacteria | | | | | | | |
| 2 | - | 10 | 2 | 10 | 1 | 5 | 5 |
| Bacteroidota | | | | | | | |
| 3 | 1 | 4 | 4 | 4 | - | - | 1 |
| Verrucomicrobia | | | | | | | |
| - | - | - | - | 2 | - | - | - |
| Campylobacterota | | | | | | | |
| - | - | - | - | - | - | 2 | 1 |
| Desulfobacterota | | | | | | | |
| - | - | - | 2 | - | - | - | 3 |
| Desulfuromonadota | | | | | | | |
| - | - | - | 1 | - | - | - | - |
| Acidobacteriota | | | | | | | |
| - | - | - | 1 | - | - | - | - |
| Spirochaetota | | | | | | | |
| - | - | - | 1 | - | - | - | - |
| No ID | | | | | | | |
| - | 1 | 0 | 2 | - | - | - | 1 |
| Crenarchaeota | | | | | | | |
| - | 1 | - | - | - | - | - | - |

level, and 1 to the class level. Five bins that were resolved only to the domain level (Bacteria) were classified as "Bins" and not as "MAGs" (Data Set S1, Sheet 2).

The 63 MAGs belong to diverse microbial phyla, and they were distributed across the five host species and tissue types (Fig. 2). In some cases, bacterial taxa from kelp blade tissues were most closely related to bacteria collected from the rhizome or sediment of a seagrass, suggesting that closely related bacterial taxa can associate with diverse hosts. Anaerobic sulfur cycling microbes, like *Desulfobulbia*, *Desulfobacteria*, *Desulfuromonadia,* and *Campylobacteria* (*Sulfurovum sp000296775* and *Sulfurimonas autotrophica*), were exclusively found in the low-oxygen rhizome and sediment environments of *Zostera marina* and *Phyllospadix* spp. Conversely, *Alphaproteobacteria* were exclusively found on surfaces exposed to the water column, while *Gammaproteobacteria* was the only class found across the range of tissue types (6 out of 8 host environments) (Data Set S1, Sheet 2).

**Host-associated microbial genomes contain pathways to synthesize vitamins, recycle nitrogen and sulfur, and use host-generated carbon.** Using the Network Algorithm for Metabolism Detection (NAMeD; described in Materials and Methods), we found evidence for several metabolic pathways that are likely important for exchanges between macrophyte hosts and their microbial partners (Fig. 3). Microbes on hosts had genes for diverse carbohydrate and carboxylic acid assimilation via cell membrane transport proteins. Host-associated microbes also had genes for a diversity of nitrogen

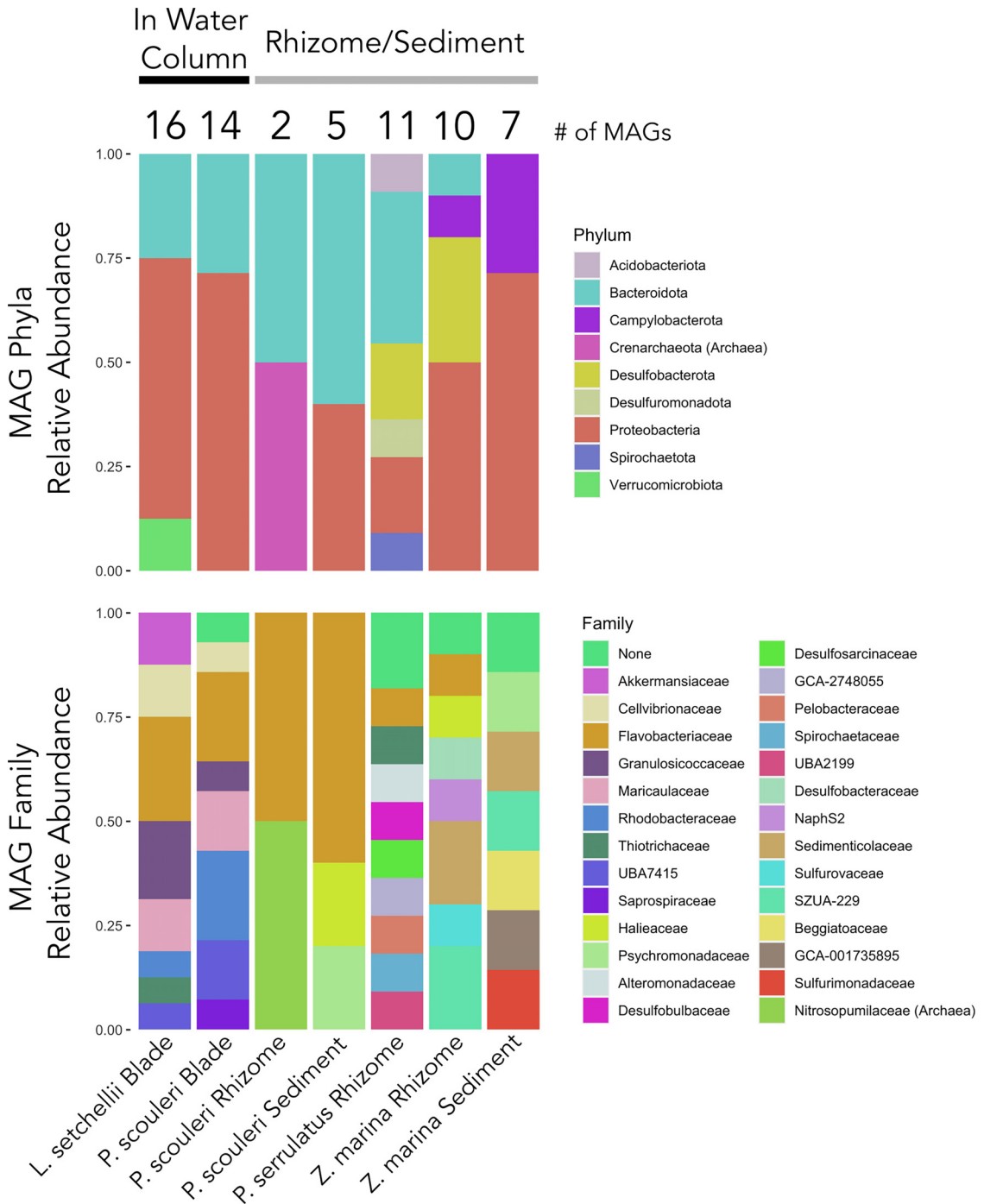

**FIG 2** Stacked bar plots showing the relative abundances of 63 metagenome-assembled genomes (MAGs) across 7 environmental samples, including 30 high quality and 33 medium quality MAGs. Only 1 MAG (*UBA7415 sp002470515*, a *Gammaproteobacteria*) was found in the bulb of *N. luetkeana* and is not displayed here.

metabolisms, including ureases and hydrolases that could generate ammonium. Nitrogen metabolisms were most diverse in the rhizome and sediment samples where we identified nitrogen recycling metabolisms (dissimilatory nitrate reduction, nitrogen fixation, denitrification). Conversely, assimilatory nitrate assimilation occurred exclusively on the surface of tissues directly exposed to the water column.

Every host sample had at least one gene from B-vitamin biosynthesis pathways (Fig. 3, see supplemental code at https://github.com/kkmiranda/PNWMetagenomes). We determined

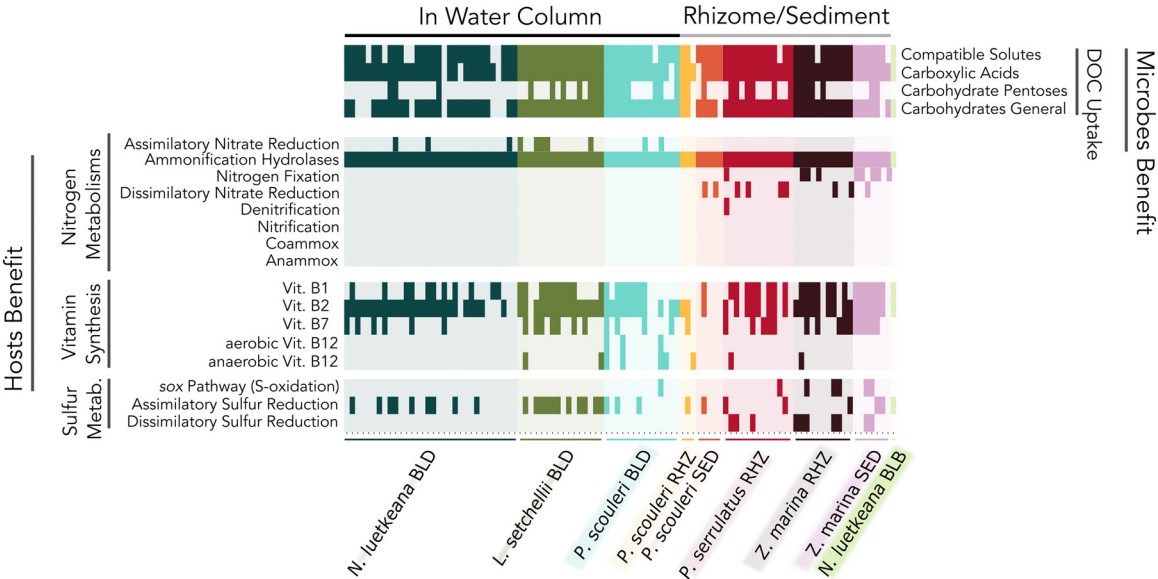

**FIG 3** Boolean heatmap of microbial metabolisms in the MAGs reported in Fig. 2 and in Data Set S1, Sheet 2, across all hosts, grouped as those that might benefit the host ("hosts benefit") and microbial metabolisms that might use host provisioned metabolites ("microbes benefit"). Each tick along the *x*-axis corresponds to a MAG. The *N. luetkeana* blade MAGs are from Weigel et al. (2022). The suite of genes for DOC Uptake (Compatible Solutes, Carboxylic Acids, Carbohydrate Pentoses, and General Carbohydrates) and Ammonification Hydrolases are shown as present if any of their genes are found in our MAGs. The presence or absence of metabolisms related to Nitrogen and Sulfur Metabolisms and Vitamin Synthesis are determined by the Network Algorithm for Metabolism Detection (NAMeD) (see supplemental code at https://github.com/kkmiranda/PNWMetagenomes/tree/main/NAMeD). The suite of genes and metabolic pathways used in this analysis can be found in Data Set S1, Sheet 3. A figure with a detailed *x*-axis is provided in Fig. S2.

that all hosts had at least one MAG with the metabolic pathway to synthesize vitamins B$_1$ (with the exception of the *P. scouleri* rhizome), B$_2$, and B$_7$ (except inside the bulb of *N. luetkeana*). The Vitamin B12 anaerobic biosynthesis pathway, however, was only present in MAGs found on the blades of *L. setchellii* (2) and *P. scouleri* (4) and the rhizomes of *P. scouleri* (1), *P. serrulatus* (1), and *Z. marina* (1). Additionally, three MAGs on the blade of *P. scouleri* that had this anaerobic pathway had the genes necessary to synthesize Vitamin B12 aerobically as well.

**Novel detection of *nifH* genes in surfgrass.** We identified the nitrogenase gene (*nifH*) in 9 MAGs with E value support $< 1.3e\text{-}120$ (KEGG) and $< 1.1e\text{-}135$ (COG). These 9 MAGs were assembled from the *P. serrulatus* rhizome (*n* = 1), *Z. marina* rhizomes (*n* = 3), and the surrounding sediment (*n* = 5). Of these 9 MAGs, 5 were resolved to the genus level (*Aliagavorans*, *Sulfurimonas*, *Thiodiazotropha*, *Eudorea*, and *Sulfurovum*), while others were resolved to the order and family levels, including *Campylobacterales*, *Desulfobacterales*, and 2 *Flavobacteriaceae* (Fig. 4, and Data Set S1, Sheet 4). The *nifH* genes from the rhizomes of *P. serrulatus* and *Z. marina* belonging to the class *Desulfobacteria* and the family *Flavobacteriaceae* belong to Cluster III: anaerobic nitrogen-fixers that are often coupled with sulfate-reduction (Fig. 4). The *Z. marina* sediment and rhizome samples also contained 3 *Campylobacteria* that had *nifH* genes that clustered together in a sister clade to the aerobic nitrogen-fixers of Cluster I.

We note that the COG gene identified as *nifH* (COG1348) also includes the homologous protochlorophyllides, which are involved in photosynthetic pigment synthesis but have high sequence similarity to the *nifH* gene (34, 47). Instead, we used the KEGG gene (K02588) that does not match these homologs. To further confirm the presence of nitrogen fixation genes in these MAGs, we inspected genes on the same contig with *nifH* and found several genes related to nitrogen fixation (Data Set S1, Sheet 4). These additional genes included *nifD* (COG 2710) in 6 of the 9 contigs, nitrogen regulatory protein PII (COG 347), *nifB* (COG 535), and multiple iron containing proteins, including ferrodoxin and Fe-Mo cluster-binding proteins (Data Set S1, Sheet 4).

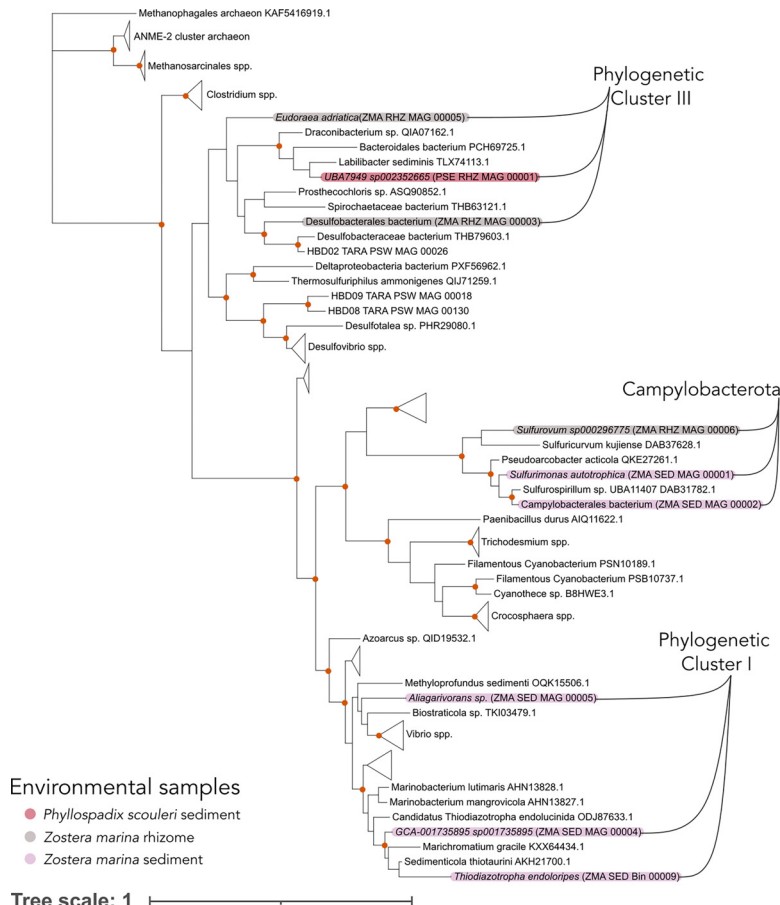

**FIG 4** A phylogenetic tree of the *nifH* genes found on the rhizomes of *P. serrulatus* (PSE, *n* = 1) and the rhizomes and surrounding sediment of *Z. marina* (ZMA, *n* = 3 and 5, respectively). Some *nifH* genes group into Cluster I, including a sulfur oxidizing taxon on the rhizome of *Z. marina* and other taxa in *Campylobacterota*, including *Sulfurovum*. Cluster III contains taxa associated with rhizomes, including *Desulfobulbus mediterraneus* on *P. serrulatus* and a *Desulfobacterales* associated with *Z. marina* rhizomes. A bootstrap support of >90% is indicated by an orange dot at a branch fork. A detailed figure with uncollapsed clades is available in Fig. S3.

## DISCUSSION

**Macrophytes and their rhizospheres host distinct microbial taxa.** The macrophyte species we sampled in this study are foundational in coastal ecosystems (4, 48–50), yet little is known about the diversity and function of their microbiomes. All MAGs were bacterial, except for a single archaeal MAG (*Crenarchaeota*) in the rhizome of *Phyllospadix scouleri*, which was identified as *Nitrosopumulis*, a genus associated with nitrification (Data Set S1, Sheet 2). Together, these 5 macrophytes hosted bacteria from 9 phyla. Blades of kelp and surfgrass were a locus of microbial diversity and function, a finding that is similar to many recent studies of macroalgal and seagrass microbiomes reporting high microbial diversity (6, 8, 9, 51–53). The functional attributes of microbial taxa associated with marine macrophytes include pathogen resistance (54), the ability to provision the host with B vitamins (11), and enhanced host algal fitness (55), perhaps through some of the nitrogen or sulfur metabolisms we documented here (5, 13).

The only low-diversity sample was the interior of the bulb of *Nereocystis*, where we assembled only a single *Gammaproteobacterial* MAG (*UBA7415 sp002470515*), suggesting that this environment of high carbon monoxide and nitrogen gas (42) may inhibit microbial activity or pose a highly selective environment. The bulb is the oldest section of the individual and functions to keep the macrophyte elevated at the surface for access to light. Unlike the blade, which may be detached at no great expense to the individual, any structural damage to the bulb may result in changes to the drag forces

experienced by whole individual, which could cause detachment and mortality (56). An area of future investigation is whether the gases within the bulb, in addition to providing buoyancy, also inhibit microbial growth.

**Host-microbe interactions in a dynamic oxygen microenvironment.** Grouping MAGs by oxygen environment (Fig. 3) showed key functional differences among macrophyte hosts, likely due to the large differences in oxygen in time and space. The 3-fold differences in dissolved oxygen (DO) between the rhizosphere and the surrounding seawater were observed at near-peak photoperiod (Fig. 1). During the night, when respiration dominates, DO in the seawater of rocky shore tide pools drops to 2 mg L$^{-1}$ (20), and the rhizomes are likely anoxic. While the oxygen environment may have little influence on ammonium production from ammonification hydrolases, blade tissues that were in contact with the oxygenated water column exclusively exhibited assimilatory nitrate reduction. The low oxygen rhizosphere, conversely, is an area where anaerobic nitrogen recycling metabolisms, including denitrification, dissimilatory nitrate reduction, and nitrogen fixation are possible. Ammonification, DNRA, and nitrogen fixation all may facilitate the availability of ammonium for host uptake. Interestingly, *UBA7947 sp002352665* (PSE_RHZ_MAG_00001) in the rhizome of *P. serrulatus* had genes for both nitrogen fixation and denitrification, both of which are metabolisms that convert nitrogen gas to ammonium and vice-versa. This suggests a variable response to the nitrogen environment, as this microbe may be denitrifying during high ammonium availability and fixing nitrogen gas when ammonium is scarce. Genes for nitrification were present, though no genes in the pathway were ever complete in any MAG. The archaeon *Nitrosopumulis* sp. from the blades of *P. scouleri* surprisingly lacked nitrification genes, despite the previous demonstration that it is capable of nitrification (57).

The abundance of dissolved organic carbon from macrophytes (21–23, 48) might select for heterotrophic metabolisms. Indeed, we found an abundance of genes for dissolved organic matter assimilation and transport in all metagenomes, suggesting that heterotrophy may be common in macrophyte-associated microbial communities (43). Improved characterization of the components of dissolved organic matter and the genomes of hosts will allow us to better assess the complementarity in resource supply by hosts and resource use by microbes, with critical implications for the carbon cycle (58).

The host tissue types in this study differed in surface oxygen concentrations. Kelp and surfgrass blade tissues interact with the water column and are likely more oxygenated than rhizome tissues or sediments, though a previous study suggests that there can also be a 60% reduction in oxygen along the surface of kelp blades during respiration (59). The thick surface mucus layer, where some kelp-associated bacteria reside (10), may further create oxygen microenvironments for aerobic or anaerobic microbial processes. Over two-thirds of the bacterial taxa on the blades of *N. luetkeana* belonged to families associated with obligately aerobic metabolisms, demonstrating the role of oxygen in shaping macrophyte-associated microbial communities (60). In contrast, the sediment surrounding the rhizomes of *Phyllospadix* spp. was a low oxygen microenvironment, likely maintained by macroinvertebrate respiration (16, 61). Low rhizosphere oxygen concentrations likely structured the taxonomic composition of *Z. marina* to include anaerobic taxa such as *Campylobacteria*, *Desulfatitalea*, and *Desulfobulbus*. The presence of anaerobes like *Desulfuromonadia*, *Desulfobacteria*, *Spirochaeta*, and *Aminicenantia* in *P. serrulatus* rhizomes suggests that sulfate reduction also occurs, possibly coupled to the use of dissolved organic carbon as an energy source (62). Additionally, *Campylobacteria* and the genus *Thiodiazotropha* were associated with *Z. marina* and may remove detrimental sulfide accumulation through sulfur oxidation (63, 64).

We encountered a range of oxidizing and reducing sulfur metabolisms. As expected, assimilatory sulfate reduction was found across all samples. Dissimilatory sulfate reduction, however, was found exclusively in rhizome/sediment conditions where oxygen is less available as a terminal electron acceptor. The product of this metabolism is hydrogen sulfide, which is toxic to eukaryotic cells. However, it also contains a large amount of chemical energy which can be released through oxidation, often through the *sox*-oxidation

pathway. Thus, microbes, like the two *Gammaproteobacteria* (ZMA_RHZ_MAG_00010, ZMA_SED_MAG_00004), two *Sedimenticolaceae* (ZMA_RHZ_MAG_00004, ZMA_RHZ_ MAG_00009), and the *Beggiatoales* (ZMA_SED_MAG_00003) present in the *Zostera* sample, may have both sulfur oxidizing and reducing metabolisms (25). In the low $O_2$ rhizome of *P. serrulatus*, *Leucothrix* sp. may couple *sox*-oxidation with dissimilatory nitrate reduction, where nitrate is used as the terminal electron receptor (25). Sulfur oxidation may stimulate microbial and host productivity by replenishing the available stock of sulfate, which serves as both a nutrient and a terminal electron acceptor (65).

We detected evidence for biosynthetic pathways for vitamins $B_1$, $B_2$, $B_7$, and $B_{12}$ that are required by the auxotrophic hosts in this study (11, 12, 66). Only the blades of *P. scouleri* harbored MAGs with both anaerobic and aerobic biosynthetic pathways for Vitamin $B_{12}$, suggesting that the variable oxygen environment driven by host metabolism creates diverse pathways for vitamin biosynthesis. While we did not detect a complete $B_{12}$ biosynthesis pathway in the *N. luetkeana* blade sample included in this study, Weigel et al. (2022) found all 22 genes necessary for the complete synthesis of $B_{12}$ in a *Granulosicoccus* MAG from *N. luetkeana* sampled in a different year, suggesting further investigation into the extent to which microbes provide this vitamin to their macrophyte hosts.

**Characteristics of previously undescribed nitrogen fixation in surfgrass.** Building on recent studies that illustrate the association of nitrogen fixing microbes with a diversity of macroalgae (67) and seagrasses (35, 36, 68, 69), we detected *nifBHDK* genes in a *Flavobacteriaceae* from *P. serrulatus* rhizomes. The *nifH* gene resolved into the Cluster III group of *nifH*, characterized by anaerobic nitrogen fixers. *P. serrulatus*, in comparison to *P. scouleri*, is found higher up in the intertidal zone and is often in sheltered tidepools that tend to undergo dramatic daily fluctuations in oxygen, possibly allowing for a temporal low-$O_2$ niche during the night (Fig. 4) (20, 70). Conversely, we did not detect nitrogenase genes in the microbiome of *P. scouleri*, which inhabits more wave-exposed, and thus, better oxygenated, environments (Fig. 1). However, stable isotope analyses from the *P. scouleri* samples show a lower nitrogen isotopic signature in the rhizome compared to the rest of the plant, a possible indication of nitrogen from an atmospheric source (Fig. S1), though in situ experiments with stable isotope tracers are needed to confirm the presence of nitrogen fixation.

Nitrogen fixation by microbial associates provides a key means of increasing the availability of ammonium, possibly supporting primary productivity. *P. scouleri* biomass reaches 12.7 kg of wet mass per square meter of shore and exudes 0.93 mg C per hour, per gram of dry mass, as dissolved organic carbon (48), which may fuel microbial activity. There is evidence that nitrogen fixation can contribute to seagrass productivity (5, 71), a possible adaptation to low nitrogen environments. Our finding that nitrogen fixing microbes are associated with a rocky intertidal surfgrass is especially surprising, given that Tatoosh Island is in an area of upwelling and high DIN at the more northerly end of the California Current Large Marine Ecosystem (72). Whether nitrogen fixation forms the basis for a reciprocal host-microbe exchange is still unknown.

The metagenomic analyses we present here suggest that macrophyte-associated microbiomes may be involved in carbon, nitrogen, sulfur, and vitamin metabolisms important to their hosts, likely generating commensal or mutualistic interactions. Our findings echo those found for marine animals, in which strong gradients in oxygen in corals and sponges are associated with diverse microbial metabolisms (73–75). Fluctuating oxygen microenvironments may promote cross-feeding, where microbial taxa produce intermediate metabolites that can be further processed by other taxa, further contributing to nitrogen (76) and carbon metabolisms (77, 78) in microbial communities. Future experiments should explore the spatial extent of these metabolisms and test hypothesized interactions between the host, its microbiome, and carbon and nitrogen cycling. The importance of seaweeds and seagrasses to coastal productivity, and the demonstrated sensitivity of both the host and microbes to increasing temperatures and pH (14, 38, 52), pathogens (51), and other anthropogenic stressors, underline the importance of further studies of host-microbiome interactions in these foundational species.

## MATERIALS AND METHODS

**The study system.** The rocky shores of the northeast Pacific Ocean are a mosaic of seagrasses and kelp (48, 79), and they include both understory and canopy kelp species. These macrophytes interact with the water column and are also anchored directly to the substrate through attachment structures known as holdfasts and rhizomes. In the case of the surfgrasses *P. scouleri* and *P. serrulatus*, the rhizomes trap sediment, creating a unique environment around the basal parts of the surfgrasses (61). Tatoosh Island, WA, is an area of high wave energy and high macrophyte diversity (80). Although typically a well-oxygenated environment, we hypothesized that the biological activities of species in wave-influenced coastal systems can create microenvironments low in oxygen. We sampled a range of kelp and seagrass species and used metagenomic sequencing and microbial genome assembly to describe the microbial taxa and metabolisms present.

**Quantifying the oxygen environment.** We quantified the oxygen microenvironment surrounding *Phyllospadix* spp. rhizomes by comparing the dissolved oxygen (DO) concentrations in the surrounding seawater and in the sediment around the rhizome. We used a Pyro Science Robust Oxygen Probe (OXROB10, Firesting, Pyroscience), and repeated measurements around 0900 h across 4 days (7 to 9 June 2019, 13 June 2021) within *P. scouleri* ($n = 18$) and *P. serrulatus* ($n = 11$) rhizomes. Each reading first measured the surrounding seawater, after which we gently pushed the tip of the oxygen probe into the sediment and rhizome mass to a depth of 1 to 3 mm, the typical thickness of the surrounding sediment on this rocky substrate (personal observation). We let the probe equilibrate and took a reading after 150 seconds. This allowed the rhizome oxygen environment to equilibrate after we disturbed the intact rhizome. We compared surrounding water and within-rhizome oxygen using paired *t*-tests in R.

**Sampling and DNA extraction.** We collected metagenome samples from distinct surfaces of 5 different macrophyte species that we hypothesized differed in their oxygen environment (Data Set S1, Sheet 1). The surfaces of *Phyllospadix scouleri* blades and *Laminaria setchellii* fronds interact with a well-oxygenated water column. We swabbed their blade surfaces with a sterile swab and brushed them with an interdental brush (GUM Proxabrush Go-Betweens) to fully census all microbes adhered to the surface, including those in the mucus layer (10). We sampled the relatively low-oxygen environment of the inner bulbs of *Nereocystis luetkeana* in the same way.

We hypothesized that seagrass rhizomes and the associated sediment would be low-oxygen microenvironments in these high-energy rocky shores. We preserved 2 cm sections of the rhizomes of *Phyllospadix scouleri*, *P. serrulatus*, and *Zostera marina*, rinsing each with sterile fresh water, and enclosing it in a sterile vial. The sediment surrounding *P. scouleri* and *Z. marina* was carefully collected with an ethanol-rinsed metal spatula and was deposited in a sterile vial. All samples were collected from Tatoosh Island, WA, USA (48.393679, −124.734617), on 16 to 17 Jul 2019, except for the *Z. marina* samples, which were sampled from West Falmouth Bay, MA, USA (41.60708333, −70.64527778), on 19 Sept 2019. We included samples from the rhizosphere of *Z. marina* from the Atlantic Ocean as a known positive control for nitrogen fixation (35, 36). Swabs, tissue, and sediment were immediately frozen at 20°C and shipped to storage at −80°C. DNA was extracted from tissue pieces approximately 1 cm in length. We used a Qiagen PowerSoil Kit, following all kit methodologies. We pooled multiple individual sample extractions for each metagenome sample to increase the DNA quantity and possible discovery: *P. scouleri* blade, rhizome and sediment (3 pooled individuals each), *P. serrulatus* rhizome (3 individuals), *L. setchellii* blade (3 individuals), *N. luetkeana* interior bulb (4 individuals), and *Z. marina* rhizomes and sediment (2 individuals).

**Shotgun metagenomic sequencing, assembly, and read recruitment.** The above 8 samples were run over 2 lanes on a HiSeq 2500 (2 × 150) with TruSeq DNA library preps at Argonne National Laboratory. For each sample, resulting DNA sequences were first quality filtered (81), then assembled with IDBA-UD v1.1.3 (82) with a minimum scaffold length of 1 kbp. Metagenomic short reads from each sample were then recruited back to their corresponding assembled contigs using Bowtie2 (83). Samtools (84) was used to generate sorted and indexed BAM files. Anvi'o v7.0 (85) was used as the command line environment for all downstream analyses. The 'anvi-gen-contigs-database' was used to generate anvi'o contigs databases, during which Prodigal v2.6.3 (86) identified open reading frames, and 'anvi-run-hmms' was used to identify genes matching to archaeal and bacterial single-copy core gene collections using HMMER (87).

**Reconstructing metagenome-assembled genomes (MAGs).** To reconstruct genomes from the assembled metagenomes, we used a combination of automatic binning via CONCOCT v1.1.0 (88), followed by a manual curation of each MAG, as outlined by Shaiber et al. 2020 (89). Genome taxonomy was determined using GTDB v.1.3.0 (90) and 'anvi-run-scg-taxonomy'. We also inferred gene-level taxonomy using Centrifuge v1.0.4 (91) to aid in the manual curation.

**Functional analysis of microbial communities.** To address the metabolic capabilities of host-associated microbes, we annotated genes in each anvi'o contigs database using 'anvi-run-kegg-kofams' and 'anvi-run-ncbi-cogs', which used the databases of the Kyoto Encyclopedia of Genes and Genomes (KEGG) (92) and NCBI's Clusters of Orthologous Genes (COGs) (93), respectively. We used these annotated genes to test for: (i) nitrogen cycling metabolisms, especially those within the nitrogen-fixation pathway, (ii) vitamin production, namely, vitamins $B_1$, $B_2$, $B_7$, and $B_{12}$, (iii) sulfur cycling metabolisms, (iv) ammonification hydrolases (EC:1.4.*, EC:3.5.*, EC:4.3.1.*), including ureases and ammonia-lyases that cleave the C-N bonds in amino acids and make ammonium available to the host, and (v) a set of dissolved organic matter (DOM) transporter genes, identified by Poretsky et al. (94), that indicate the ability of the microbial community to assimilate DOM exudates from kelps and surfgrasses. The presence or absence of these metabolisms are displayed in a Boolean heatmap in Fig. 3. For (i), (ii), and (iii), we developed and used a reproducible algorithm on KEGG definitions to detect the presence of

these biosynthetic pathways, calling it the Network Algorithm for Metabolism Detection (NAMeD) (see supplementary code at https://github.com/kkmiranda/PNWMetagenomes/tree/main/NAMeD). Given that we were searching for metabolisms in novel environments where some genes might be poorly documented, we set the criteria in NAMeD for determining the presence of a metabolism at 67% of the genes present in a given pathway. For (iv) and (v), we assembled suites of genes which we searched for within each MAG (suites are described in Data Set S1, Sheet 3). To expand our functional analysis of the kelp microbiome, we included 32 MAGs from the surface of *N. luetkeana* blades (43), derived from metagenome samples TAT_2019a and TAT_2019b, collected from Tatoosh Island at the same time as the samples in our study.

**Phylogenetic analysis of nifH genes.** To search for *nifH* amino acid sequences in our environmental samples, we identified 9 MAGs which contained *both nifH* and *nifD* genes using the KEGG identifiers K02588 and K02586 with E values of <1e-100. We then aligned these 9 *nifH* AA sequences against 89 well-characterized reference *nifH* AA sequences (Data Set S1, Sheet 5) using Muscle v3.8.1 (95) and refined the alignment using trimAl (gap-threshold: 0.5) (96) and 'anvi-script-reformat-fasta' (max-percentage-gap: 50%). A maximum-likelihood phylogeny was inferred using IQTree (97) with 1,000 bootstrap replicates, and a LG+R5 model best fit our data, found using ModelFinder (98). The *nifH* genes from the *Zostera* samples served as positive controls with which to detect nitrogen fixation genes in other samples. Fig. 2 to 4 were generated using iTol v5 (99) and R v4.0.3 (100) and were edited in Inkscape. Finally, we sampled tissues from *P. scouleri* rhizomes ($n = 16$), the basal meristematic region just distal to the sheath ($n = 12$), and blades 35 cm above the rhizome ($n = 12$) to quantify stable isotopes of $\delta^{15}N$ and $\delta^{13}C$ to look for signatures of nitrogen fixation (methods described in Text S1, results displayed in Fig. S1).

**Data availability.** In addition to the code available on github.com/kkmiranda/PNWMetagenomes, the final MAG database files generated in anvi'o are available on the FigShare repository: https://doi.org/10.6084/m9.figshare.20152949.v3. Metagenomic sequence data are available at the NCBI's Sequence Read Archive under accession number PRJNA813168.

## SUPPLEMENTAL MATERIAL

Supplemental material is available online only.
**DATA SET S1**, XLSX file, 0.1 MB.
**TEXT S1**, DOCX file, 0.1 MB.
**FIG S1**, TIFF file, 13 MB.
**FIG S2**, TIFF file, 68 MB.
**FIG S3**, TIFF file, 9 MB.

## ACKNOWLEDGMENTS

Our gratitude is to the Makah Tribal Nation for access to Tatoosh Island. We thank The University of Chicago's Microbiome Center for pilot award funding, Washington Department of Natural Resources, grants 93099282 and 93100399 (CAP), and NSF-DEB grant (no. 1556874) awarded to J.T. Wootton. We appreciate the work of C. Sauceda in the isotope analysis, as well as A. Wootton, A. Wood, and K. Foreman in the field sampling. S. Owens and S. Greenwald at Argonne National Lab provided expertise in sequencing. K.M. was supported by an EE Fellowship from The University of Chicago.

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
