## [Reviewer comments · mSystems]

The diversity and functional capacity of microbes associated with coastal macrophytes

Khashiff Miranda, Brooke Weigel, Emily Fogarty, Iva Veseli, A. Murat Eren, Anne Giblin, and Catherine Pfister

Corresponding Author(s): Khashiff Miranda, Université Laval

Review Timeline:

Submission Date:

July 2, 2022

Accepted:

July 7, 2022

Editor: Christopher Anderton

Reviewer(s): The reviewers have opted to remain anonymous.

Transaction Report:

DOI: <https://doi.org/10.1128/msystems.00592-22>

July 7, 2022

Mx. Khashiff K Miranda
Université Laval
Biology
Québec, Québec G1V 0A6
Canada

Re: mSystems00592-22 (The diversity and functional capacity of microbes associated with coastal macrophytes)

Dear Mx. Khashiff K Miranda:

Your manuscript has been accepted, and I am forwarding it to the ASM Journals Department for publication. For your reference, ASM Journals' address is given below. Before it can be scheduled for publication, your manuscript will be checked by the mSystems production staff to make sure that all elements meet the technical requirements for publication. They will contact you if anything needs to be revised before copyediting and production can begin. Otherwise, you will be notified when your proofs are ready to be viewed.

Publication Fees:

If you would like to submit a potential Featured Image, please email a file and a short legend to mssystems@asmusa.org. Please note that we can only consider images that (i) the authors created or own and (ii) have not been previously published. By submitting, you agree that the image can be used under the same terms as the published article. File requirements: square dimensions (4" x 4"), 300 dpi resolution, RGB colorspace, TIF file format.

We recognize that the video files can become quite large, and so to avoid quality loss ASM suggests sending the video file via <https://www.wetransfer.com/>. When you have a final version of the video and the still ready to share, please send it to mSystems staff at mssystems@asmusa.org.

Sincerely,

Christopher Anderton
Editor, mSystems

Journals Department
Appendix 1: Accept
Dataset S1: Accept
Fig. S2: Accept
Fig. S1: Accept
Fig. S3: Accept